# Molecular Detection and Phylogenetic Analyses of *Babesia* spp. and *Theileria* spp. in Livestock in Bangladesh

**DOI:** 10.3390/microorganisms11061563

**Published:** 2023-06-13

**Authors:** Uday Kumar Mohanta, Boniface Chikufenji, Eloiza May Galon, Shengwei Ji, Zhuowei Ma, Shimaa Abd El-Salam El-Sayed, Aaron Edmond Ringo, Thanh Thom Do, Xuenan Xuan

**Affiliations:** 1National Research Center for Protozoan Diseases, Obihiro University of Agriculture and Veterinary Medicine, Obihiro 080-8555, Hokkaido, Japan; ukmohanta.mipa@sau.edu.bd (U.K.M.); preciouschikufenji@gmail.com (B.C.); eloizagalon@gmail.com (E.M.G.); jishengwei0903@hotmail.com (S.J.); mazhuowei1994@gmail.com (Z.M.); shimaa_a@mans.edu.eg (S.A.E.-S.E.-S.); aringo2002@yahoo.com (A.E.R.); thanhthomdo@gmail.com (T.T.D.); 2Department of Microbiology and Parasitology, Sher-e-Bangla Agricultural University, Sher-e-Bangla Nagar, Dhaka 1207, Bangladesh

**Keywords:** *Babesia* spp., *Theileria* spp., molecular detection, phylogeny, livestock, Bangladesh

## Abstract

Piroplasmosis, caused by *Babesia* spp. and *Theileria* spp., poses significant constraints for livestock production and upgradation in Bangladesh. Besides examining blood smears, few molecular reports are available from some selected areas in the country. Therefore, the actual scenario of piroplasmosis in Bangladesh is deficient. This study aimed to screen the piroplasms in different livestock species by molecular tools. A total of 276 blood samples were collected from cattle (*Bos indicus*), gayals (*Bos frontalis*) and goats (*Capra hircus*) in five geographies of Bangladesh. After that, screening was conducted through a polymerase chain reaction, and species were confirmed by sequencing. The prevalence of *Babesia bigemina*, *B. bovis*, *B. naoakii*, *B. ovis*, *Theileria annulata* and *T. orientalis* was 49.28%, 0.72%, 1.09%, 32.26%, 6.52% and 46.01%, respectively. The highest prevalence (79/109; 72.48%) of co-infections was observed with *B. bigemina* and *T. orientalis*. The phylogenetic analyses revealed that the sequences of *B. bigemina* (*BbigRAP-1a*), *B. bovis* (*BboSBP-4*), *B. naoakii* (*AMA-1*), *B. ovis* (ssu rRNA) and *T. annulata* (*Tams-1*) were included in one clade in the respective phylograms. In contrast, *T. orientalis* (MPSP) sequences were separated into two clades, corresponding to Types 5 and 7. To our knowledge, this is the first molecular report on piroplasms in gayals and goats in Bangladesh.

## 1. Introduction

Piroplasms, *Babesia* spp. and *Theileria* spp., are tick-borne protozoan parasites causing piroplasmosis in livestock and wild animals worldwide [1]. The species of the genus *Babesia* infect red blood cells (RBC), while *Theileria* spp. infect both RBC and white blood cells. However, the transmission of piroplasms occurs by different tick species of the family Ixodidae [2]. The economic impact on the livestock industry by piroplasmosis may be due to a loss of production, reduced working efficiency, cost of treatment and prevention, and morbidity and mortality [3,4].

Bovine clinical babesiosis, caused by *B. bigemina*, *B. bovis*, *B. naoakii* and *B. divergens*, leads to intravascular haemolytic anaemia [5,6] in cattle and small ruminants. Among these *Babesia* spp., *B. bovis* is described as the most pathogenic species and causes neurological and respiratory disorders in animals, leading to death. On the other hand, several other *Babesia* spp., *B. ovata*, *B. major* and *B. occultans*, are known to be less pathogenic and cause subclinical infections. In contrast, *B. ovis* is reported as the most pathogenic for sheep and goats in tropical and subtropical areas and is characterized by fever, anaemia, icterus, haemoglobinuria and death [7]. Babesiosis is mainly transmitted by *Rhipicephalus* (*Boophilus*) *microplus*.

On the other hand, clinical theileriosis is caused by *T. parva* and *T. annulata*, which are transmitted by *Hyalomma* spp. *Theileria parva* causes East Coast Fever (ECF) in eastern, central and southern Africa, whereas *T. annulata* causes tropical (Mediterranean) theileriosis in North Africa, southern Europe and Asia [8]. Moreover, *T. mutans*, *T. tautoragi* and *T. velifera* are also reported to cause bovine theileriosis in Africa [9,10]. *Theileria annulata* causes lymphoproliferative disease [11] manifested by fever, inappetence, lymphadenopathy, icterus, tachycardia, dyspnoea and weakness [12]. Mortality due to tropical theileriosis may range from 90% in the newly introduced exotic breed to nearly 5% in indigenous breeds [13]. Therefore, the disease threatens livestock production and improvement in developing countries [11]. On the contrary, *T. orientalis*, *T. sergenti* and *T. buffali* are regarded as non-lymphoproliferative benign theileriosis [14], infecting a wide variety of hosts globally [15]. Although *T. orientalis* is considered benign, disease outbreak has been reported in several countries [16,17,18,19,20].

Bangladesh is an agriculture-based developing country in South Asia. It lies between 23°41′39.52″ N and 90°20′39.67″ E. Livestock being an integral part of agriculture, it contributes 1.90% to gross domestic products (GDP) and 16.52% to agricultural products [21]. The country has 24.7 million cattle head, 1.5 million buffaloes, 26.7 million goats and 3.7 million sheep. The livestock sector provides 20% full-time and 50% partial employment to the rural people. Along with the fast-growing economy of the country, the demand for livestock products and by-products is also increasing. Unfortunately, tick-borne diseases (TBDs), particularly piroplasmosis, pose a significant threat to animal upgradation programs. The tick species previously reported from Bangladesh are *R. microplus*, *R. sanguineus*, *Hyalomma anatolicum*, *Haemaphysalis bispinosa* and *Amblyomma testudinarium* [22,23,24]. In contrast, tick-borne pathogens (TBPs) reported so far are bovine babesiosis, theileriosis and anaplasmosis from some selected areas. Most of these reports are based on thin blood smears [25,26,27,28,29], while a few are based on molecular tools from selected areas [30,31]. Therefore, a complete scenario on piroplasms in different livestock species from all over the country is deficient. In this study, blood samples from three animal species in five different locations were analysed to screen for piroplasms.

## 2. Materials and Methods

### 2.1. Ethics Statement

Approval for animal sampling was obtained from the Department of Livestock Services (DLS), Ministry of Fisheries and Livestock, Government of the People’s Republic of Bangladesh. Verbal consent was attained from the animals’ owners through providing them with detailed study objectives. Blood samples were collected by registered veterinarians through proper restraining to avoid any injury to the animals. Moreover, the ethical guidelines for the use of animal samples approved by Obihiro University of Agriculture and Veterinary Medicine, Obihiro, Hokkaido 080-8555, Japan (Animal experiment approval ID number: 22-23) were followed during sample collection.

### 2.2. Study Sites and Sample Collection

From June 2021 to March 2022, blood samples were collected from cattle (*Bos indicus*), gayals (*Bos frontalis*) and goats (*Capra hircus*) in five different areas of Bangladesh, namely the Jhenaidah, Bogura, Sirajganj and Bandarban districts and Naikhongchari upazila (sub-district) (Figure 1). The Jhenaidah district was selected because of its border with India in the West. Bogura was chosen as a representative of the northern districts. The Sirajganj district was selected for its unique livestock rearing practices called Bathan, acres of fallow grassland where animals are housed and maintained in the dry season (December–June). The Bandarban district and the Naikhongchari sub-district were selected as representatives of hill districts. Although Naikhongchari is a sub-district of Bandarban, it was considered separately because it is bordered by Myanmar in the southeast and a coastal district, Cox’s Bazar, in the West. Blood samples were collected randomly from 276 apparently healthy animals, namely cattle (*Bos indicus*; 174), gayal *(Bos frontalis*; 9) and goats (*Capra hircus*; 93) in Jhenaidah (29), Bogura (14), Sirajganj (107), Bandarban (105) and Naikhongchari (21). Approximately 2 mL of blood from the jugular vein of each animal was collected in an EDTA-coated vacutainer tube (BD Bioscience, Bergen County, NJ, USA). All three species of sampled animals were categorized into two groups, namely young (<2 years old) and adult (≥2 years old). The age of the animals was confirmed through dentition of the animals and farmers’ record books. The collected blood samples were stored in a cool box in the field and refrigerator (4 °C) in the laboratory.

### 2.3. Dried Blood Spot Preparation on FTA^TM^ Elute Micro Card and DNA Elution

A total of 30 μL of blood was withdrawn from the collection tube and dispensed in a concentric circular motion on one of the circles of Whatman FTA^TM^ elute micro cards (GE Healthcare, Buckinghamshire, UK). Four circles of one FTA^TM^ elute micro card were loaded with blood samples from different animals. The loaded cards were allowed to dry thoroughly for at least three hours and stored at room temperature until DNA elution. Genomic DNA was eluted from the loaded circles of the FTA^TM^ elute micro cards following the manufacturer’s guidelines. In brief, the cards were placed on a cutting mat (2.5″ × 3.0″), and a 3 mm disc was punched out from each of the circles using a UniCore punch kit (QIAGEN, Hilden, Germany) and placed in 1.5 mL sterile microcentrifuge tube. The disc was then rinsed in 500 μL sterile water by vortexing three times for five seconds, and the water was removed using a sterile pipette. The washed disc was then centrifuged at 11,000 rpm for 30 s, and excess water was removed. After adding 50 μL sterile water, the tubes containing washed discs were placed in a heating block at 98 °C for 30 min. Following heat treatment, the samples were vortexed and centrifuged at 11,000 rpm for 30 s to separate the disc from the eluate. Finally, the discs were removed from the tube, and the eluted DNA was stored at −30 °C.

### 2.4. Molecular Detection of Piroplasms

Genomic DNA was screened for *Babesia* spp. (*B. bigemina*, *B. bovis*, *B. naoakii* and *B. ovis*) and *Theileria* spp. (*T. annulata* and *T. orientalis*) by species-specific polymerase chain reaction (PCR) assays (Table 1). Nested PCR (nPCR) assays were only used for *B. bigemina*, *B. bovis* and *T. annulata*. Notably, *B. ovis* was only screened from goat blood samples. DNA fragments of the target genes were amplified by PCR in a total volume of 10 μL, containing 0.05 μL One *Taq* DNA Polymerase (New England Biolabs, Ipswich, UK), 0.2 μL of deoxynucleotide triphosphate mix (dNTPs; 10 mM), 0.2 μL of each primer (10 μM), 2.0 μL of 5X One *Taq* standard buffer, 1 μL template DNA and 6.35 µL of UltraPure™ DNase-/RNase-free distilled water (Invitrogen, Waltham, MA, USA). The thermal cycle conditions for the PCR essays in this study were adopted from previous studies [7,11,32,33,34]. The amplified PCR products were then subjected to electrophoresis in a 1.5% agarose gel, stained with ethidium bromide, and visualized under UV-transilluminator.

### 2.5. Sequencing of the PCR-Positive Samples

For sequencing, 2–7 PCR-positive samples from each pathogen were randomly selected. The PCR amplicons were extracted from the gel and purified by using the Nucleo Spin^®^ Gel and PCR Clean-up kit (Macherey Nagel, Düren, Germany). The concentration of the extracted products was measured by the NanoDrop 2000 spectrophotometer (ThermoFisher Scientific, Waltham, MA, USA). Sequence reactions were performed using the BigDye™ Terminator v3.1 Cycle Sequencing Kit (Applied Biosystems, Foster City, CA, USA) and ABI Prism 3100 Genetic Analyzer (Applied Biosystems). The resultant sequences were trimmed and assembled using the CodonCode Aligner version 9 (CodonCode Corporation, Centerville, MA, USA) to obtain consensus sequences. GenBank BLASTn analysis was conducted to confirm the identity of the sequences to those already registered in the GenBank database. Shared percent identities among the sequences of each pathogen were calculated through EMBL-EBI Clustal Omega multiple sequence alignment.

### 2.6. Phylogeny Construction

Phylogenetic analyses of the sequences of *B. bigemina* (*BbigRAP-1a*), *B. bovis* (*BboSBP-4*), *B. naoakii* (*AMA-1*), *B. ovis* (ssu rRNA), *T. annulata* (*Tams-1*) and *T. orientalis* (*MPSP*) were conducted by the maximum likelihood (ML) method using MEGA XI [35]. The ML method was used to select the best nucleotide substitution model based on a Bayesian information criterion for ML analyses, and the phylogeny test was carried out by the bootstrap method with 1000 replications.

### 2.7. GenBank Accession Numbers

Accession numbers for the sequences generated in this study were obtained through depositing in the GenBank database of the National Center for Biotechnology Information (NCBI), using BankIt for the genomic DNA sequences and the ribosomal RNA submission portal (submit.ncbi.nlm.nih.gov/subs/genbank/; (accessed on 27 December 2022)) for the ribosomal RNA sequences. Assigned accession numbers for the sequences generated in this study are OQ162124–OQ162130 (*B. bigemina*), OQ144958 and OQ144959 (*B. bovis*), OQ148404 and OQ148405 (*B. naoakii*), OQ130581 and OQ130582 (*B. ovis*), OQ162131–OQ162136 (*T. annulata*) and OQ144960–OQ144964 (*T. orientalis*).

### 2.8. Statistical Analyses

The data comprised locations (Jhenaidah, Bogura, Sirajganj, Bandarban and Naikhongchari), and animal parameters (age, gender and species) were considered independent variables. Data for low detection rates were excluded from the analyses. Pearson’s chi-square and Fisher’s exact test were used to assess the association of TBP detection rates in different study locations, age, sex and species of the animals in GraphPad Prism 8 (GraphPad Software, San Diego, CA, USA). A *p*-value was considered significant when it was <0.05.

## 3. Results

### 3.1. Overall Prevalence

Among the 276 animals screened for *Babesia* spp. and *Theileria* spp. by PCR assays, 203 animals (73.55%) were positive for at least one of the six piroplasms, while 73 animals (26.45%) were not infected by any of the piroplasms. The piroplasms detected in this study are *B. bigemina* (49.28%), *B. bovis* (0.72%), *B. naoakii* (1.09%), *B. ovis* (32.26%), *T. annulata* (6.52%) and *T. orientalis* (46.01%) (Table 2). A significant difference in the prevalence of *B. bigemina* and *T. orientalis* was detected among the study locations (Table 2). A geographic separation was observed in the prevalence of *B. naoakii* (Sirajganj district) and *T. annulata* (Badarban district and Naikhongchari sub-district). The prevalence of *B. bigemina* (54.60%) and *T. orientalis* (67.24%) was significantly higher in cattle than that in goats (Table 3). With respect to the age groups, a significantly higher prevalence of *T. orientalis* was observed in adult goats (≥2 years old) (Table 3). On the other hand, no significant difference in the prevalence was observed between the sexes of animals (Appendix A).

### 3.2. Co-Infections with Different Piroplasms

The animals infected with two or more pathogens were considered co-infections. Among the 276 animals, 109 animals (39.49%) were found to be infected with two or more piroplasms (Appendix A). The number of piroplasms in co-infections ranged from double to triple. Infections with double pathogens contributed the majority of co-infections (104/109; 95.41%), whereas triple infections were observed only in 4.59% of the animals (5/109). The most recurrent combinations of co-infections were with *B. bigemina* and *T. orientalis* (79/109; 72.48%), followed by *B. bigemina* and *B. ovis* (12/109; 11.0%) and *T. annulata* and *T. orientalis* (7/109; 6.42%). Among the study locations, the highest prevalence of co-infections was observed in Bandarban (51/105; 48.57%), followed by Naikhongchari (8/21; 38.10%), Sirajganj (40/107; 37.38%), Bogura (4/14; 28.57%) and Jhenaidah (6/29; 20.69%). No significant difference in the prevalence of co-infections was observed among the study locations.

### 3.3. Gene Sequence Analyses

The identity for the sequences of each detected pathogen was compared among themselves and with the reference sequences in the GenBank. The percent nucleotide identities of seven *BbigRAP-1a* (*B. bigemina*; 412 bp) sequences (OQ162124–OQ162130) in this study ranged from 99.76–100%. Additionally, these sequences shared 99.75–100% identity values with MH790974 (Bangladesh) and MG210824 (Tanzania). The shared nucleotide percent identity value between the two isolates of *B. bovis* (*BboSBP-4*; 500 bp) generated in this study (OQ144958 and OQ144959) was 99.40%. These two isolates of *B. bovis* shared 95.53–95.96% identity values with ON012668 (Kenya) and ON012677 (Australia). In the case of *B. naoakii* (*AMA-1*; 373 bp), two sequences (OQ148404 and OQ148405) generated in this study shared a percent identity of 99.73% between them, while these two sequences shared a 99.19–99.46% identity with the isolate from Sri Lanka (LC385894). Furthermore, the ssu rRNA sequences of *B. ovis* (OQ130581 and OQ130582; 550 bp) displayed a percent identity of 96.58% with each other. Interestingly, they showed a higher identity of 99.45–99.64% and 99.41–99.61% with OP003548 (the Philippines) and KF723611 (Tunisia), respectively. All six sequences of *T. annulata* (*Tams-1*; 453 bp) in this study were identical. However, these sequences shared a 99.56% identity with the sequences from Turkey (AF214914) and Egypt (MZ197898, MN251046 and MN251047). On the contrary, the percent identity among the sequences (OQ144960–OQ144964) of *T. orientalis* (776 bp) ranged from 82.22–99.61%. Nevertheless, these sequences shared an identity of 99.87% with LC438477 (Sri Lanka) and 100% with AB560818 (Viet Nam).

### 3.4. Phylogenetic Analyses

The phylogenetic trees of *B. bigemina* (*BbigRAP-1a*), *B. bovis* (*BboSBP-4*), *B. naoakii* (*AMA-1*), *B. ovis* (ssu rRNA), *T. annulata* (*Tams-1*) and *T. orientalis* (*MPSP*) were constructed along with their respective reference sequences from the GenBank database. In the phylogram, all the sequences of *B. bigemina* (*BbigRAP-1a*) clustered together in the same clade and displayed a close relationship to those reported previously from Bangladesh (MH790974), Tanzania (MG210824), South Africa (MK481015), Benin (KX685380) and Burkina Faso (OK323209) (Figure 2). The phylogram constructed from the *BboSBP-4* gene sequences of *B. bovis* showed that the isolates from Bangladesh were clustered in one clade along with those from Australia (ON012677) and Kenya (ON012668) (Figure 3). Moreover, *B. naoakii* sequences (*AMA-1*) in this study were included in the same clade and had a close phylogenetic relationship with LC385894 (Sri Lanka) and LC486015 (Viet Nam) (Figure 4). Similarly, *B. ovis* (ssu rRNA) isolates from Bangladesh formed a monophyletic clade with those from Tunisia (KF23611 and KP670199), the Philippines (OP003548), Iran (KY581550 and KY581551), Iraq (MN309742 and MN560046), Turkey (KY867435 and MG569902), Bosnia and Herzegovina (OM758310), Spain (AY533146) and Portugal (KJ829366) (Appendix A). However, all *T. annulata* isolates formed a monophyletic clade which was a sister to the clade formed by isolates from Egypt (MN251046, MN251047 and MZ197898) and Turkey (AF214914) (Appendix A). On the contrary, the *T orientalis* (*MPSP*) isolates from Bangladesh were distributed in two clades in the phylogram, representing two genotypes, 5 and 7 (Figure 5).

## 4. Discussion

This study was conducted in wide geographic areas, eight agro-ecological zones of Bangladesh [36], including two predominant livestock species (cattle and goats) and one nearly endangered species (gayals) [37]. Moreover, this is the first study on the detection of tick-borne pathogens from livestock in Jhenaidah, Bogura and Bandarban. In addition, to our knowledge, this study provides the first molecular data on the piroplasms in goats and gayals from Bangladesh.

This study observed a very high prevalence (73.55%) of piroplasms in livestock from Bangladesh. Among the piroplasms, *B. bigemina*, *B. bovis*, *B. naoakii*, *B. ovis*, *T. annulata* and *T. orientalis* were found to infect the animals screened. Moreover, a high rate of co-infections with two or more pathogens was also detected in this study. Among the four *Babesia* spp. screened, the prevalence of *B. bigemina* was the highest (49.27%), followed by *B. ovis* (32.26%), *B. naoakii* (1.09%) and *B. bovis* (0.72%). Although the prevalence of *B. bovis* is supported by a previous report [30] (0.5%), the prevalence of *B. bigemina* and *B. naoakii* was much lower in previous studies [30,31]. The lower prevalence in these previous studies might be due to the inclusion of samples from limited areas and the selection of crossbred cattle [31]. Crossbred cattle are mainly managed in an intensive system (except for the Sirajganj district) and therefore do not receive exposure to ticks in the pasture. A relatively high prevalence of *B. ovis* (32.26%) was observed in goats. Although Black Bengal goats are famous worldwide for their meat and skin, they are always neglected to be monitored for TBPs. In this study, the existence of *B. ovis* in goats in Bangladesh was confirmed by sequencing for the first time. As *B. ovis* has been regarded as the most pathogenic for sheep [7], a detailed study on the pathogenicity of this piroplasm in Bangladesh is urgently needed. Among the different study locations, the prevalence of *B. bigemina* was significantly higher in Sirajganj (61.68%) and Bandarban (48.57%). Moreover, *B. bovis* was only found in Sirajganj and Bandarban, while *B. naoakii* was only detected in Sirajganj. In addition, the detection rate of *B. ovis* was also higher (37.09%) in Sirajganj. In relation to the livestock species, the prevalence of *B. bigemina* was significantly higher in cattle (54.60%). The higher prevalence of *B. bigemina* in cattle might be due to the rearing system, semi-intensive in Bandarban and extensive in Sirajganj in the dry season, exposing them to vector ticks, *R*. (*Boophilus*) *microplus*. The phylogenetic analyses suggest that each of *Babesia* spp. formed a monophyletic clade with other reference sequences in the respective phylograms, indicating that a single genotype for each species of *Babesia* is circulating in Bangladesh.

Two *Theileria* spp., *T. annulata* and *T orientalis*, were screened from the livestock in this study. The prevalence of *T. annulata* in this study was supported by previous reports [25,31]. Among the different study areas, *T. annulata* was only detected in the hill tracts (Bandarban and Naikhongchari). Hilly areas are rich in vegetation where the vector ticks, *Haemaphysalis* spp., are abundant. In addition, semi-intensive rearing practices of livestock in the hills allow them to be exposed to vectors more frequently than in other parts of the country. On the other hand, *T. orientalis* was found as the third most abundant (46.01%) species, which is in line with the findings of a previous study [30]. The prevalence of *T. orientalis* is significantly higher in Bandarban (68.57%) and Sirajganj (29.90%), which might be due to the higher exposure of livestock to vector, *H. bispinosa* (semi-intensive in Bandarban and an extensive system in dry seasons in Sirajganj). The detection rate of *T. orientalis* was significantly higher in cattle (67.24%) than in caprine (7.53%). The lower prevalence of *T. orientalis* in goats could be due to their intensive management system. Moreover, a significantly higher prevalence of *T. orientalis* was observed in goats over two years old. The lower prevalence in young goats might be associated with the acquired immunity from the mother and/or less exposure to tick vectors. These findings in this study were supported by a phenomenon called inverse age resistance [38,39,40]. Inverse age resistance in young animals is developed through acquiring specific immunoglobulin (especially immunoglobulin G) from the mother via colostrum, which confers innate immunity [41]. In the phylogram, the isolates of *T. annulata* from Bangladesh were identical and formed a monophyletic clade with the isolates from Egypt and Turkey. On the contrary, the isolates of *T. orientalis* were distributed into two different clades, corresponding to Types 5 and 7, in the phylogram constructed from the *MPSP* gene. Among the 11 different genotypes of *T. orientalis*, Type 2 [20] and Type 7 [17] are reported to be associated with clinical cases. Nevertheless, two genotypes, including Type 7, were detected among the five sequences generated in this study. Therefore, a considerable number of sequences from all the study areas may provide precise data on the circulating genotypes of *T. orientalis* in Bangladesh.

A very high prevalence of co-infections (39.49%) was observed among the animals examined, which is supported by previous reports [30]. The high prevalence of co-infections with different piroplasms is associated with the distribution of tick vectors in the country. *Ripicephalus* (*Boophilus*) *microplus* and *H. bispinosa* are the most abundant tick species throughout Bangladesh [23,42]. These tick species are the predominant vectors for piroplasms in Bangladesh. Therefore, the prevalence of multiple infections is high in livestock in the country. Among the study locations, the highest prevalence of co-infections was detected in the hills (48.57% in Bandarban and 38.10% in Naikhongchari), which might be associated with the highest prevalence of *Ripicephalus* (*Boophilus*) *microplus*, *H. bispinosa* and *A. testudinarium* (only prevalent in the hills) in the hills [23].

In general, almost all of the piroplasms screened in this study were highly prevalent in Sirajganj and Bandarban. The high prevalence in Sirajganj might be due to the Bathan system. On the other hand, Bandarban is a hilly area where the humidity, rainfall and vegetation are higher than in other parts of the country, providing suitable conditions for breeding different tick species. In the hills, animals are allowed to graze freely throughout the day, further allowing them to be infested by ticks. A geographic separation was observed in *B. naoakii* (only in Sirajganj) and *T. annulata* (Bandarban and Naikhongchari). Although the exact reason behind the geographic separation of piroplasms is still unknown, proper distribution mapping may aid in preventing and controlling these piroplasms in Bangladesh.

## 5. Conclusions

This study investigates the molecular prevalence of piroplasms in goats, the second most important livestock species, and gayals (besides cattle) from Bangladesh for the first time. Moreover, this is the first report on piroplasms from livestock in the studied areas, except for Sirajganj and Naikhongchari. In addition, a previously reported piroplasm, *B ovis*, was confirmed here by sequencing for the first time from Bangladesh. Therefore, our study focuses on the necessity for further investigation on the piroplasms in Bangladesh. However, the inclusion of more samples from other geographies may provide a better picture of the piroplasms of livestock in the country for formulating future control and prevention strategies.

## Figures and Tables

**Figure 1 microorganisms-11-01563-f001:**
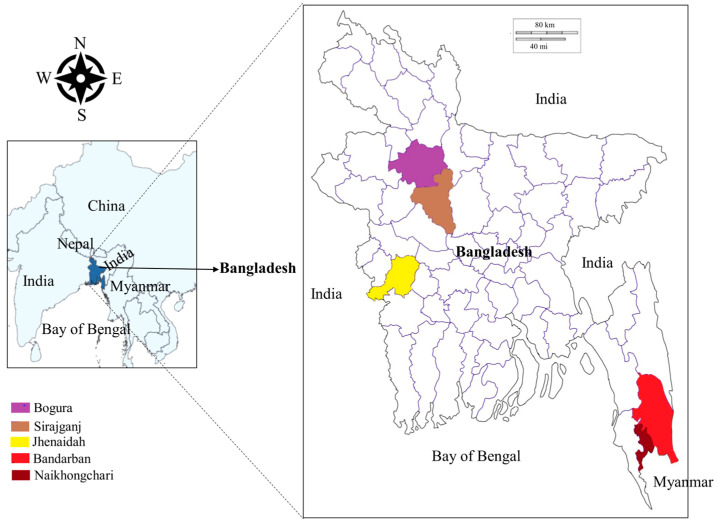
Map of Bangladesh highlighting sampling locations.

**Figure 2 microorganisms-11-01563-f002:**
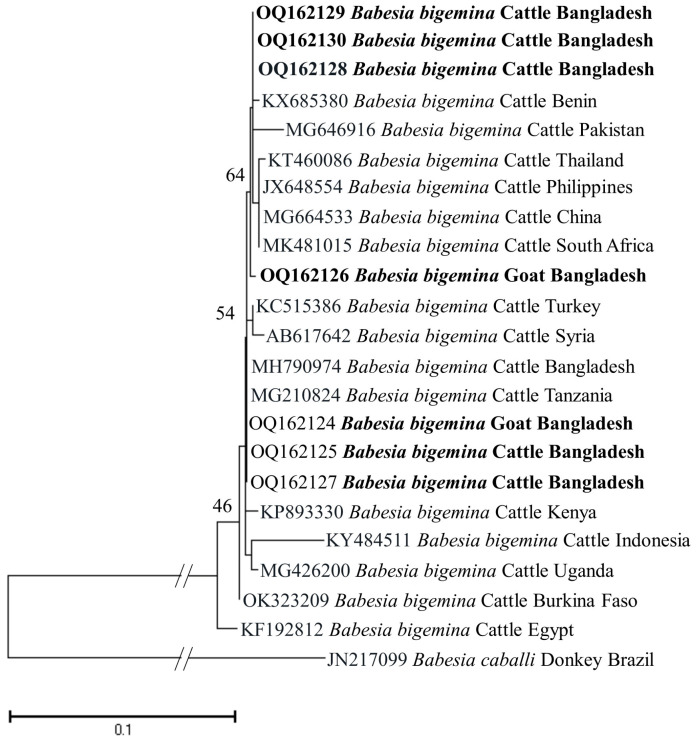
A maximum likelihood phylogram of *B. bigemina* inferred from *BbigRAP-1a* gene. The tree was constructed by MEGA X using the Kimura 2-parameter model. The sequences of this study are shown in boldface. The *RAP-1a* gene sequence of *B. caballi* (JN217099) was used as an outgroup. Numbers on the nodes indicate the percentage of 1000 bootstrap replicates. The scale bar indicates the estimated number of nucleotide substitutions per position.

**Figure 3 microorganisms-11-01563-f003:**
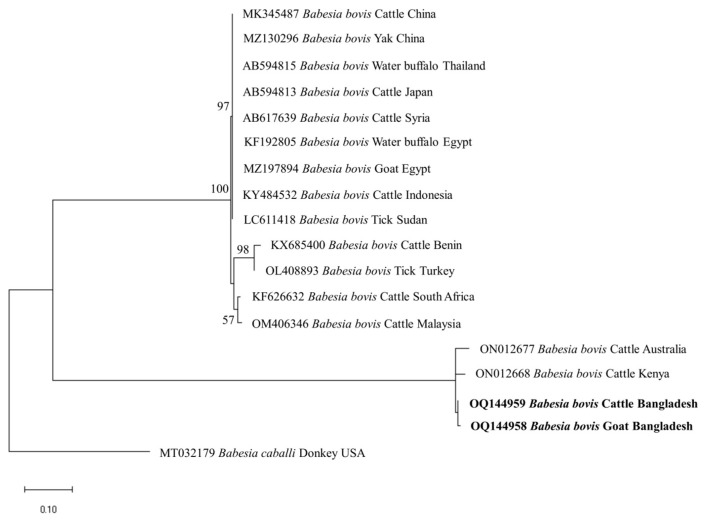
A maximum likelihood phylogram of *B. bovis* inferred from *BbovSBP-4* gene. The tree was constructed by MEGA X using the Kimura 2-parameter model. The sequences of this study are shown in boldface. The *SBP-4* gene sequence of *B. caballi* (MT032179) was used as an outgroup. Numbers on the nodes indicate the percentage of 1000 bootstrap replicates. The scale bar indicates the estimated number of nucleotide substitutions per position.

**Figure 4 microorganisms-11-01563-f004:**
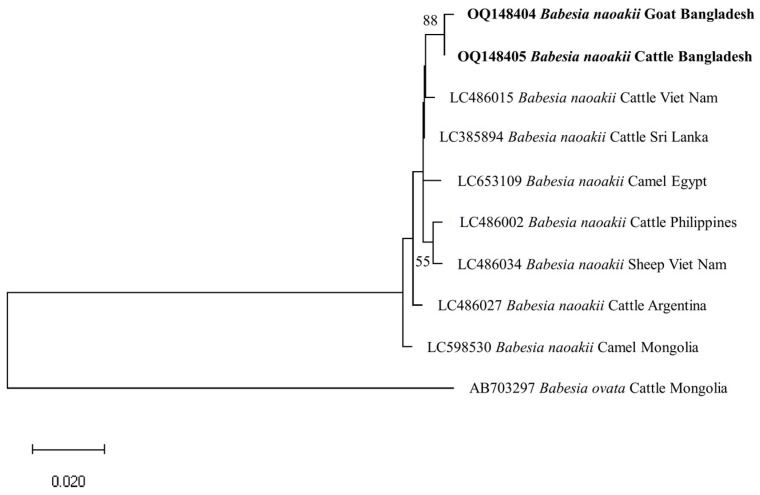
A maximum likelihood phylogram of *B. naoakii* inferred from *AMA-1* gene. The tree was constructed by MEGA X using the Tamura 3-parameter model. The sequences of this study are shown in boldface. The *AMA-1* gene sequence of *B. ovata* (AB703297) was used as an outgroup. Numbers on the nodes indicate the percentage of 1000 bootstrap replicates. The scale bar indicates the estimated number of nucleotide substitutions per position.

**Figure 5 microorganisms-11-01563-f005:**
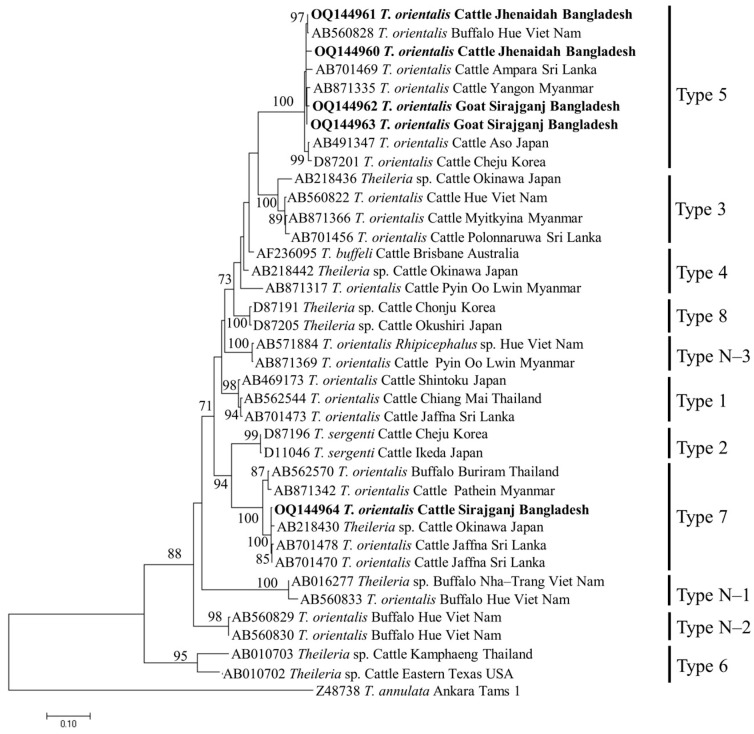
A maximum likelihood phylogram of *T. orientalis* inferred from *MPSP* gene. The tree was constructed by MEGA X using the Tamura–Nei model with a discrete Gamma distribution. The sequences of this study are shown in boldface. The *Tams1* sequence of *T. annulata* (Z48738) was used as an outgroup. Numbers on the nodes indicate the percentage of 1000 bootstrap replicates.

**Table 1 microorganisms-11-01563-t001:** List of primers used for the detection of piroplasms.

Target Gene	Assays	Primer Sequences Forward 5′ → 3′ Reverse		Annealing Temp. (°C)	References
*B. bigemina* (*BbigRAP-1a*)	PCR	GAGTCTGCCAAATCCTTAC	TCCTCTACAGCTGCTTCG	55	[34]
	nPCR	AGCTTGCTTTCACAACTCGCC	TTGGTGCTTTGACCGACGACAT	50	
*B. bovis* (*BboSBP-4*)	PCR	AGTTGTTGGAGGAGGCTAAT	TCCTTCTCGGCGTCCTTTTC	55	[34]
	nPCR	GAAATCCCTGTTCCAGAG	TCGTTGATAACACTGCAA	50	
*B. naoakii* (*AMA-1*)	PCR	TGGCGCCGACTTCCTGGAGCCCATCTCCAA	AGCTGGGGCCCTCCTTCGATGAACCGTCGG	64	[32]
*B. ovis* (ssu rRNA)	PCR	TGGGCAGGACCTTGGTTCTTCT	CCGCGTAGCGCCGGCTAAATA	62	[7]
*T. annulata* (*Tams-1*)	PCR	GTAACCTTTAAAAACGT	GTTACGAACATGGGTTT	54	[11]
	nPCR	CACCTCAACATACCCC	TGACCCACTTATCGTCC	54	
*T. orientalis* (*MPSP*)	PCR	CTTTGCCTAGGATACTTCCT	ACGGCAAGTGGTGAGAACT	58	[33]

**Table 2 microorganisms-11-01563-t002:** Geographic distribution of piroplasms in livestock of Bangladesh.

Pathogens	Locations					Total n = 276	*p*-Value
	Jhenaidah n = 29	Bogura n = 14	Sirajganj n = 107	Bandarban n = 105	Naikhongchari n = 21		
*B. bigemina*	9 (31.03%)	8 (57.14%)	66 (61.68%)	51 (48.57%)	2 (9.52%)	136 (49.28%)	<0.001
*B. bovis*	n.d.	n.d.	1 (0.93%)	1 (0.95%)	n.d.	2 (0.72%)	NA
*B. naoakii*	n.d.	n.d.	3 (2.80%)	n.d.	n.d.	3 (1.09%)	NA
*B. ovis* *	6 (27.27%)	n.d.	23 (37.09%)	1 (14.29%)	n.d.	30 (32.26%)	NA
*T. annulata*	n.d.	n.d.	n.d.	4 (3.81%)	14 (66.67%)	18 (6.52%)	NA
*T. orientalis*	7 (24.13%)	7 (50%)	32 (29.90%)	72 (68.57%)	9 (42.86%)	127 (46.01%)	<0.001

*: only screened for goats (n = 93); NA: not analysed; n.d.: not detected.

**Table 3 microorganisms-11-01563-t003:** Prevalence of piroplasms according to the species and age of the host animals in Bangladesh.

Pathogens	Animal Species									Total (n = 276)
	Cattle (n = 174)			Gayals (n = 9)			Goats (n = 93)			
	<2 yrs (n = 64)	≥2 yrs (n = 110)	Sub Total	<2 yrs (n = 5)	≥2 yrs (n = 4)	Sub Total	<2 yrs (n = 63)	≥2 yrs (n = 30)	Sub Total	
*B. bigemina*	40 (22.99%)	55 (50.00%)	95 (54.60%) *	1 (20.00%)	n.d.	1 (11.11%) *	25 (39.68%)	15 (50.00%)	40 (43.01%) *	136 (49.28%)
*B. bovis*	1 (0.57%)	n.d.	1 (0.57%)	n.d.	n.d.	n.d.	1 (1.59%)	n.d.	1 (1.08%)	2 (0.72%)
*B. naoakii*	1 (0.57%)	n.d.	1 (0.57%)	n.d.	n.d.	n.d.	n.d.	2 (6.67%)	2 (2.15%)	3 (1.09%)
B. ovis †	n.s.	n.s.	n.s.	n.d.	n.s.	n.s.	24 (38.10%)	6 (20.00%)	30 (32.26%)	30 (32.26%)
*T. annulata*	4 (2.30)	5 (4.55%)	9 (5.17%)	5 (100.00%)	2 (50.00%)	7 (77.78%)	1 (1.59%)	1 (3.33%)	2 (2.15%)	18 (6.52%)
*T. orientalis*	46 (26.44%)	71 (64.55%)	117 (67.24%) **	3 (60.00%)	1 (25.00%)	4 (44.44%) **	2 (3.17%) *	5 (16.67%) *	7 (7.53%) **	128 (46.01%)

†: only screened for goats; n.d.: not detected; n.s.: not screened; *****: *p* ˂ 0.05; **: *p* ˂ 0.001.

## Data Availability

The datasets generated during and/or analysed during the current study are available from the corresponding author on reasonable request.

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
