# Peer review of "Molecular Detection and Phylogenetic Analyses of Babesia spp. and Theileria spp. in Livestock in Bangladesh"

_microorganisms, 2023, doi:10.3390/microorganisms11061563_

Round 1

Reviewer 1 Report

This study by Mohanta et al describes the detection of multiple species of Babesia and Theileria in farmed ungulates from Bangladesh.  I find the study well done and well-written. My only comments are on the figures.  See below.

1.  There are just too many phylogenetic trees.  Instead of having a tree for each species detected, have one larger Babesia/Theileria tree that will confirm the identity of the parasites detected.  All the individual trees, I would put into a supplemental file, except the T. orientalis tree, which is quite interesting.

2. Similarly, there are too many tables.  Put the tables that sort the data by sex and age into the supplemental data.  While interesting, they are not crucial to the paper.

The english is quite good.  

Reviewer 2 Report

The manuscript has good potential to be published on microorganisms. However, it is essential to clarify and clear up found findings that are not mentioned throughout the study. On the other hand, points are mentioned that are evident by the presence of ticks and tick-borne diseases.

The presentation of results is not clear. Many of the tables presented arach other; I suggest only placing the most representative ones. Nothing is ever mentioned about why them; For example: if the highest prevalence in adult goats is due to T. orientalis, what would be the possible explanation? I suggest developing the results better since there is much outstanding information.

It would be important to mention the conditions in each location to elucidate the species of ticks that inhabit each region and the possible transmission of diseases to animals. Furthermore, it would be convenient to add a taxonomic detection of the vectors found in each species to attribute the presence of diseases. Because otherwise, mentioning that B. bigemina is transmitted by R. microplus is nothing new… This happens in all tropical regions of the world.

Line 373-374 This possible explanation is feeble. In this case, I expect a strong argument of why adult animals present a higher prevalence of T. orientalis.

Line 287-389 This finding is not novel. It is happening in all tropics and subtropics areas.

Reviewer 3 Report

Major Concerns:

In some cases, the number of animals tested or parasites detected is too low to draw a conclusion.  For example, only 9 gayals were tested across the 5 geographic locations and statements about a lower prevalence of certain parasites in gayals should be eliminated. (Line s199 and 370).  Similarly, there were only 2 detections of B. bovis and these were in the locations where the highest number of animals were tested.  Given such low numbers it is not possible to draw a conclusion about the geographic distribution of B. bovis (line 196).  The lack of detection in 3 of the 5 locations may be due to the low numbers of animals tested in those areas which is less than 30 compared to ~100 in the two locations where it was detected.

There are sets of primers listed as nPCR for 3 of the parasites.  To what does the nPCR refer? If nested PCR was performed it was not described in the Materials and Methods.

What controls were performed to be certain there was no cross contamination of amplified products?  This is particularly important since blood from more than one animal was collected on a single card and tubes containing amplified product were opened in order to visualize products on agarose gels.  Was unidirectional workflow used?

What is the size of the amplified products and what cutoffs were used to determine the identity of the sequenced products?  Were there any criteria used to consider the next closest sequence match?

The number of figures is high for the amount of data presented.  Suggest combining tables 3 and 4 and eliminating gayals from the table since the number of animals tested is very low. Suggest eliminating table 5 since no significant difference were found on the basis of sex.  Similarly, some of the phylogenetic relationships could be described but do not need to be included as a figure.

It is unclear how the study included eight agro-ecological zones (line 326) as there were only 5 geographic locations tested and in two cases two areas were adjacent.  Also, the acronym AEZ is not needed as the term is only used once.

The discussion and the conclusion both state that B. ovis is going to be confirmed (future tense) as if it had not been performed in this study. 

Minor Points:

Suggest cutting the line for the distance to the outgroup for B. bigemina in Figure 2 so the distances for the remaining sequences can be seen more easily.  They are currently undistinguishable.

The acronym TBP(s) is used but not defined.

Both B. bovis and B. ovis are described as being the most pathogenic (lines 45 and 48). Suggest rewording.

Suggest changing “known to be benign” to “considered benign” in line 62 since the rest of the sentence describes disease outbreaks associated with T. oreintalis.

In line 75 either the word “most” or “mainly” should be removed as they are redundant.

On line 79 the word “for” is missing – to screen for piroplasms.

Consistency with naming could be improved. At the beginning of the discussion the animals tested are referred to by their scientific name whereas the common names are primarily used in the tables and text, other than in the materials and methods.   Only gayals are referred to by the scientific name in the abstract. 

There are some areas where the language could be improved as odd phrases are used.  This is particularly noticeable in the abstract.

Round 2

Reviewer 2 Report

This manuscript was improved and have better presentation of the results. I think that all my concerns were considering in the final version. I have no more comments.

Author Response

Dear Reviewer 2

Thank you very much for your feedback.

Prof. Xuenan Xuan, DVM, PhD